# UCP2 as a Cancer Target through Energy Metabolism and Oxidative Stress Control

**DOI:** 10.3390/ijms232315077

**Published:** 2022-12-01

**Authors:** Angèle Luby, Marie-Clotilde Alves-Guerra

**Affiliations:** Université Paris Cité, Institut Cochin, INSERM, CNRS, F-75014 Paris, France

**Keywords:** uncoupling protein 2 (UCP2), mitochondria, metabolism, oxidative stress, cancer, therapies

## Abstract

Despite numerous therapies, cancer remains one of the leading causes of death worldwide due to the lack of markers for early detection and response to treatment in many patients. Technological advances in tumor screening and renewed interest in energy metabolism have allowed us to identify new cellular players in order to develop personalized treatments. Among the metabolic actors, the mitochondrial transporter uncoupling protein 2 (UCP2), whose expression is increased in many cancers, has been identified as an interesting target in tumor metabolic reprogramming. Over the past decade, a better understanding of its biochemical and physiological functions has established a role for UCP2 in (1) protecting cells from oxidative stress, (2) regulating tumor progression through changes in glycolytic, oxidative and calcium metabolism, and (3) increasing antitumor immunity in the tumor microenvironment to limit cancer development. With these pleiotropic roles, UCP2 can be considered as a potential tumor biomarker that may be interesting to target positively or negatively, depending on the type, metabolic status and stage of tumors, in combination with conventional chemotherapy or immunotherapy to control tumor development and increase response to treatment. This review provides an overview of the latest published science linking mitochondrial UCP2 activity to the tumor context.

## 1. Introduction

Cancer is today a major public health issue with a high incidence. With nearly 10 million deaths per year, it was the third leading cause of death worldwide in 2020 [1]. According to the definition of the World Health Organization: “cancer is the rapid multiplication of abnormal cells with unusual growth, which can then invade neighboring parts of the body, and migrate to other organs”. Hanahan and Weinberg have deepened this definition by listing all the common characteristics of cancer cells through periodic updates of several reviews [2,3,4]. After the addition of cellular energetic dysregulations to the “hallmarks of cancer”, the metabolic reprogramming of tumors has gained renewed interest [3]. In the last decade, the study of tumor metabolism, through its mechanisms, signaling pathways and metabolites, has even become a crucial issue for the understanding and treatment of cancer. Otto Warburg’s 1920 hypothesis led for almost a century to the misconception that cancer cells were exclusively dependent on glycolysis to produce adenosine tri phosphate (ATP), and that mitochondria were damaged [5]. Nevertheless, although cancer cells adopt a preferential glycolytic metabolism following the activation of oncogenes, which stabilize under hypoxic conditions, mitochondria are functional and necessary for tumorigenesis [6]. Mitochondria are indeed the production factories of cells. Energy-intensive tumor cells take up oxygen (O_2_) and nutrients (glucose and essential amino acids) to form building blocks (proteins, lipids and nucleic acids) through mitochondrial biosynthetic pathways to sustain a high proliferation rate and promote resistance to chemotherapy [7,8,9]. However, mitochondria are the main source of reactive oxygen species (ROS) production. The oxidation of glucose by oxidative phosphorylation (OXPHOS) results in the incomplete reduction of O_2_. Electron leakage at complexes I and III of the electron transport chain (ETC) leads to the release of superoxide (O_2_^−^) molecules, which are highly unstable. The superoxide is then successively converted to oxygen peroxide (H_2_O_2_) and water (H_2_O) molecules by antioxidant enzymes to maintain the redox status [10]. However, an uncontrolled increase in ROS is cytotoxic and generates oxidative damage that promotes protein and lipid damage, as well as genomic DNA mutations. This genomic instability can lead to the induction of pro-tumor genes that destabilize cellular integrity [11,12]. To cope with metabolic changes, cells have developed metabolic regulators to control excessive ROS formation. In the past two decades, many studies have highlighted the antioxidant role of uncoupling proteins (UCPs) and more specifically the UCP2 protein. The limitation of ROS production by UCPs reduces cellular stress, which would contribute to improving inflammatory responses in a tumorigenic context [13].

UCPs belong to the family of mitochondrial inner membrane transporters. UCP1 was the first member of the UCP family discovered in 1976 in mammalian brown adipose tissue [14]. Homologs, including UCP2, were then identified [15,16,17]. UCP2 is the uncoupling protein with the highest homology to UCP1 (59%), but its localization and function are distinct [18]. Although its messenger RNA is ubiquitously distributed in all organs, the protein is only detected in certain tissues such as the spleen, pancreas, lung, intestine and white adipose tissue, due to the tight regulation of its translation [19,20]. Even though the highly controversial uncoupling activity has been investigated, a recent biochemical characterization of UCP2 revealed that the protein is involved in the transport of four-carbon (C4) tricarboxylic acid (TCA) metabolites to regulate pyruvate oxidation in mitochondria [21,22]. The role of UCP2 has been implicated in various physiological and pathological conditions. Indeed, through the physiological regulation of ROS [23,24,25], the modulation of metabolism [22,26] and the control of the immune system [27,28], UCP2 has been implicated in the control of autoimmune diseases [29], cardiovascular dysfunction [30], neuronal pathologies [31] and cancers [32,33,34].

Today, although intensive research has led to major advances in the understanding of the mechanisms of tumorigenesis, therapies have evolved little, even with the emergence of immunotherapy. Indeed, immunotherapy and targeted therapies offer only limited success in curing certain patients and in specific cancers [35,36]. Radiotherapy, chemotherapy and surgery remain the first-line treatments despite the development of resistance and significant side effects. However, to improve their effectiveness, the challenge is to combine treatments by delivering, for example, targeted metabolic therapies with conventional treatments or immunotherapies. UCP2, which is a protein involved in metabolic control, could thus represent a therapeutic strategy to fight against the metabolic reprogramming of tumors [37,38].

In this review, we will first characterize and deepen our knowledge of the physiological functions of UCP2. Then, we will understand how this protein, through its multiple roles, is involved in the metabolic and immune reprogramming of tumors. Finally, we will explore whether targeting UCP2 could represent a therapeutic advantage in the fight against cancer and resistance to therapies.

## 2. Role of UCP2 in the Physiological and Pathophysiological Context

Since the late 1990s, many studies have characterized the involvement of UCP2 in several physiological and pathophysiological processes, but its biological role is not clearly elucidated. Indeed, UCP2, but also its homologue UCP3, have been the subject of much controversy (Figure 1) [13,39]. This first part will allow us to review the biochemical functions attributable to UCP2 in order to better appreciate its involvement in a physiological and pathophysiological context.

### 2.1. Biochemical Function

#### 2.1.1. Uncoupling?

The strong homology between UCP1 and UCP2 initially directed research toward similar mitochondrial membrane uncoupling functions (Figure 1). Uncoupling corresponds to the dissociation of ETC oxidative phosphorylation from ATP synthesis; it causes hydrogen ion leakage from the intermembrane space to the mitochondrial matrix to dissipate proton gradient energy. The use of recombinant UCP2 expression systems in yeast and proteoliposomes showed that proton flux decreases the membrane potential in the presence of UCP2 [15,40,41]. The same phenomenon was later described in β-pancreatic cells, thymocytes and isolated kidney mitochondria [24,42,43]. However, these results have been questioned by other studies using mitochondria isolated from *Ucp2^−/−^* spleen and kidney [21]. While a thermogenic phenotype has been described as a consequence of uncoupling [44], no dysregulation of energy balance and thermoregulation was observed in *Ucp2^−/−^* mice exposed to cold [42,45]. It has been proposed that UCP2 lowers the membrane potential to decrease the ATP/ADP ratio, and thus limit mitochondrial ROS production, during successive oxidation–reduction reactions of the respiratory chain, and negatively regulates insulin secretion [24,42,43]. Indeed, the uncoupling induced by the overexpression of UCP2 in isolated pancreatic islets decreases the electrochemical gradient, which then reduces by 50% the production of ATP by complex V, preventing the exocytosis of insulin vesicles [46]. As for UCP1, these reactions are inhibited by purine nucleotides and activated by free fatty acids, glucose, retinoic acid or ROS [41]. In 2002, Etchay et al. showed that the activation of UCP2 by superoxide increased the proton conductance in a manner dependent on fatty acids, and more particularly on palmitate in isolated mitochondria from the kidney, spleen and pancreas of rats [43]. Other authors have even stated that in the absence of these activators, UCP2 could not perform this uncoupling function [31]. However, these results on the uncoupling activity of UCP2 are now highly controversial. While the characterization of the UCP1 protein showed that two histidine residues (His-145 and His-147) were essential for its protonophore activity, they are absent in UCP2 [47]. These data suggest that if UCP2 is a proton carrier, it uses a different mechanism. Moreover, studies conducted to elucidate the uncoupling role of UCP2 were performed under overexpression conditions in yeast and bacterial systems [15,40,41,48]. However, the artificial overexpression of UCP carriers in these models prevented them from reaching the correct protein conformation [13,49], leading to an uncontrolled artifactual uncoupling of mitochondrial respiration [49,50,51] that is no longer comparable to the physiological level of UCP2 adjusted by tight translation regulation [19]. Furthermore, in the basal state, a comparison of the respiratory rate of mitochondria isolated from spleen and lung of *Ucp2^+/+^* and *Ucp2^−/−^* mice showed no change [21]. Additionally, in the latter study, the activation of uncoupling with retinoic acid required micromolar concentrations, and is unlikely to be physiologically relevant [21]. These data are also in agreement with the work of Rial et al., who showed that under standard conditions, the use of purine nucleotides, palmitic acid and retinoic acid had no effect on the respiratory rate of UCP2-overexpressing yeast mitochondria [41]. In contrast, they observed that an increase in pH associated with micromolar concentrations of retinoic acid stimulated respiration through an increase in *UCP2* mRNA [41]. However, *Ucp2* is translationally regulated and an increase in mRNA expression does not necessarily correlate with protein levels [19]. Unfortunately, much of the data in favor of uncoupling activity have not been replicated. Indeed, studies have described UCP2 activity in renal tissue where UCP2 could never be detected [24,52]. Other studies have used commercial anti-UCP2 antibodies that have been shown to be nonspecific for UCP2 protein detection [42,53,54]. To address the difficulties of the interpretation of western blot, Pecqueur et al. developed in 2001 a sensitive anti-UCP2 polyclonal antibody (UCP2-605 homemade) that has since been approved [19,25]. Since then, other teams have developed antibodies against UCP2 [34]. Thus, the use of commercial antibodies must be carefully validated using *Ucp2^−/−^* tissue or UCP2^CRISPR/Cas9^ cells to ensure the specific detection of UCP2.

#### 2.1.2. Transport of Metabolites

The discovery of other UCPs in plants and fish, organisms not dependent on thermogenesis, further challenged the uncoupling theory and suggested that UCP2 may have other biochemical functions [55,56]. Indeed, UCP2 belongs to the SLC25 family, a group that includes all mitochondrial carriers [57]. The discovery in 2006 of the involvement of UCP3 in the transport of pyruvate was another argument that directed the search for UCP2 activity towards a transport function in mitochondria [58]. It was proposed by Trenker et al., using the overexpression of UCP2 in human endothelial cells, that UCP2 participates in mitochondrial calcium (Ca^2+^) sequestration [59]. However, after the discovery of mitochondrial calcium uniport (MCU), which was identified as a major contributor to mitochondrial Ca^2+^ transport, it was suggested that UCP2 might have a modulatory activity. This was confirmed by super-resolution microscopy technology, which showed a specific association between UCP2 and MCU, once the latter was methylated by the protein arginine methyl transferase 1 (PMRT1). This interaction was positively correlated with mitochondrial Ca^2+^ uptake [60,61,62]. In addition, an interaction has been demonstrated between calcium ions and TCA intermediates, such as citrate and malate [63,64]. Interestingly, Vozza and collaborators demonstrated that UCP2 catalyzes the exchange of C4 metabolites (oxaloacetate, malate, aspartate) for phosphate and hydrogen ions, and therefore regulates pyruvate oxidation in the mitochondrial matrix [22]. Thus, the TCA activity is regulated to avoid overload by the accumulation of intramitochondrial compounds and to reduce the redox pressure, and thus limit ROS production (Figure 1) [22]. These results highlight that the metabolic remodeling (regulation of oxidative stress, insulin secretion, etc.) associated with UCP2 since the early 2000s could be explained by the biochemical transport function of C4 metabolites, and not by an uncoupling activity.

### 2.2. UCP2 a Metabolic Regulator

Due to the distribution of the UCP2 protein mainly in glycolytic tissues and its role as a mitochondrial carrier, it is now well understood that UCP2 is involved in different physiological and pathological processes related to glucose and lipid metabolism, such as the regulation of food intake, insulin secretion, and immune response [65,66]. Our team has demonstrated that UCP2 acts as a metabolic rewiring regulator to favor fatty acid metabolism over glucose. Indeed, the genetic loss of *Ucp2* in murine embryonic fibroblasts (MEFs) was associated with increased proliferation correlated with glycolytic reprogramming at the expense of β-oxidation [26,67]. Indeed, in *Ucp2^+/+^* MEF cells, only 67% of ATP is produced by glucose oxidation, compared to 89% in *Ucp2^−/−^* MEF cells [68]. Glycolysis allows the supply of nucleotides through the pentose phosphate pathway to support proliferation. Subsequently, other studies supporting the hypothesis of Pecqueur et al. confirmed that UCP2 is a sensor of glucose to lipid metabolism, especially under energy stress [65]. During caloric restriction or fasting, UCP2 expression is positively correlated with an increased energy response and greater weight loss [69]. *Ucp2^−/−^* mice exposed to fasting have a predisposition to hepatic steatosis due to impaired lipid utilization [70]. These mechanisms are also related to the downregulation of insulin synthesis by UCP2 in pancreatic β-cells [71]. UCP2 thus facilitates lipolysis and consequently the resulting β-oxidation [71]. However, insulin release is ineffective after exposure to high glucose concentrations. In this case, UCP2 promotes cell deregulation by inhibiting apoptosis via bcl-2 [72]. Moreover, ghrelin is a hormone that promotes food intake during a fasting state through its activating action on NPY/AgRP neurons. UCP2 mediates the action of ghrelin by activating AMP-activated protein kinase (AMPK), which suppresses acetyl-CoA carboxylase (ACC) enzymatic activity but increases Carnitine palmitoyl transferase I (CPT1) expression to promote energy intake via β-oxidation in these neurons [73]. In contrast, Zhang et al. and Vozza et al. suggested that the loss of UCP2 promotes mitochondrial glucose oxidation by measuring the intramitochondrial accumulation of TCA constituents in stem cells and human hepatocarcinoma cells (HepG2) [22,74]. This accumulation reflects TCA saturation induced by the absence of C4 metabolite efflux. Furthermore, Kukat et al. demonstrated in mice with cardiomyopathy that the absence of the UCP2 protein prevented the oxidation of lipids accumulated in the cells [75]. ATP was then supplied by glycolysis, inducing significant lactic acidosis in disabled mice.

In addition to glucose and fatty acids, cancer and immune cells oxidize glutamine through the glutaminolysis reaction. Glutamine is a major amino acid in plasma and, as a substrate, is successively converted into glutamate and ketoglutarate to integrate the TCA and support oxidative metabolism. In the presence of glutamine, *Ucp2^−/−^* macrophages showed altered homeostasis of glutaminolysis-derived metabolites with the accumulation of glutamate, succinate and malate compared to *Ucp2^+/+^* macrophages [76]. Additionally, fatty acids and glutamine have been shown to activate *Ucp2* transcription (via Peroxisome Proliferator-activated Receptor (PPAR)) and *Ucp2* translation, respectively [77,78,79]. These data reinforce the transporter role of UCP2, which in turn can export the C4 intermediates provided by the oxidation of these two substrates independently of the presence of glucose. The export of C4 metabolites can thus renew the cellular stock of pyruvate and regulate oxidative stress in parallel via glutathione by avoiding a surplus of glucose consumption that would overload the TCA cycle in glycolytic tissues.

In conclusion, these different studies show that UCP2 is an essential regulator of substrate utilization, both at the systemic level via insulin regulation and at the cellular level.

### 2.3. UCP2 and Oxidative Stress: A Link toward Immunity

The homeostasis of redox status contributes to cellular integrity. It has been recognized that ROSs are involved in many etiologies, such as cancer, aging, neurodegeneration and inflammatory pathologies. Since the first studies on UCP2, its role as a modulator of ROS has been suggested [24,80,81]. Indeed, in 1997, Nègre-Salvayre et al. showed that when UCP2 was inhibited by GDP, mitochondrial H_2_O_2_ production was increased [80]. Subsequently, UCP2 overexpression studies in human aortic endothelial cells and neonatal rat cardiomyocytes confirmed these initial results, while emphasizing that the inhibition of ROS production by UCP2 decreased oxidative stress-induced apoptosis [82,83]. At that time, the hypothesis put forward by scientists was that the uncoupling generated by UCP2 jointly lowered mitochondrial membrane potential and ATP synthase activity. The functioning of the ETC was then reduced, which in turn decreased the formation of ROS [24,80,84]. Nowadays, in view of the findings of Vozza et al., it becomes natural to think that since UCP2 controls the oxidation of TCA substrates, the redox pressure of OXPHOS and consequently ROS production is attenuated (Figure 1) [22].

With the identification of the antioxidant role of UCP2, much research has been carried out to investigate its involvement in various pathologies, including those related to immune system activation, as evidenced by the numerous reviews recently published [85,86,87]. Hereafter, we will give a brief and non-exhaustive description of the plurality of oxidative stress responses induced by UCP2 in different pathologies.

#### 2.3.1. Insulin Regulation—Type 2 Diabetes (T2D)—Cardiovascular Diseases

Research on UCP2 quickly highlighted its role in the pathogenesis of diabetes through its regulation of pancreatic β-cell function [24]. In mouse models, UCP2 was shown to be involved in pancreatic development via the ROS-Akt (protein kinase B) pathway by promoting the proliferation of α and β endocrine cells from the embryonic stage [88], and its loss promoted glucose-stimulated insulin secretion (GSIS) compared to wild-type mice [42]. Furthermore, the short-term inhibition of UCP2 activity by Genipin, a pharmacological inhibitor of UCP2 derived from Chinese medicine, improved glycemic response, indicating that UCP2 is a negative regulator of insulin secretion [71]. These combined results may suggest a better response to hyperglycemia. Indeed, Zhang et al. demonstrated a beneficial effect of UCP2 loss on glucose homeostasis in an ob/ob diabetic mouse model [42]. These mice had better GSIS in the first phase after glucose ingestion and better insulin signal transduction in white adipose tissue [42]. However, improved insulin tolerance did not completely restore a healthy phenotype, as no significant changes in body weight, food intake, serum triglycerides, or epididymal fat were found in diabetic Sw/Uni mice treated with an antisense *Ucp2* inhibitor [89]. On the contrary, other studies showed that the reduction in *Ucp2* expression achieved by transfection of small interfering RNAs in pancreatic islets of ob/ob diabetic mice did not modify plasma insulin levels or blood glucose levels [90], but generated an imbalance of oxidative stress in the pancreatic islets [91]. The loss of UCP2, over several generations of backcrossed mice, in four different strains induced extensive macrophage immune infiltration, which subsequently caused a persistent accumulation of ROS with an imbalance in the reduced glutathione/oxidized glutathione (GSH/GSSG) ratio and an increase in nitrotyrosine levels [92,93]. This excessive accumulation of ROS therefore led to pancreatic α- and β-cells dysfunction, favoring the long-term progression to T2D [94]. Indeed, the lack of insulin feedback on glucagon led to glucose resistance. By analogy, the overexpression of UCP2 in islets conferred protection to pancreatic β-cells against oxidative stress and glucotoxicity [95]. In addition, patients with T2D have an increased risk of developing cardiovascular disease [96]. The genetic deletion of *Ucp2* in a mouse model of atherosclerosis accelerated the development of atherosclerotic lesions in the mouse aorta [30]. Cardiovascular dysfunction resulted in endothelial alterations that facilitated monocyte invasion into the intima of *Ucp2^−/−^* mice [97]. Moreover, an oxidative burst produced by *Ucp2^−/−^*-infiltrating macrophages in atherosclerotic plaques strongly and locally increased ROS, and activated pro-inflammatory cytokine release through overexpression of the pro-inflammatory transcription factor Forkhead Box Protein O1 (FoxO1) [30,98].

#### 2.3.2. Infectious Diseases

UCP2 has also been shown to be involved in antimicrobial defense by increasing ROS production after infection. Shortly after the discovery of *Ucp2*, Arsenijevic et al. demonstrated that *Ucp2^−/−^* mice were resistant to Toxoplasma gondii infection. The loss of the protein stimulates the release of ROS from macrophages, which have a toxoplasmicidal action and thus limit the formation of brain cysts [27]. After listeria infection, the improved survival of *Ucp2^−/−^* mice was also observed due to an enhanced phagocytic infiltrate following increased secretion of the MCP1 chemokine for up to 4 days [28]. At the molecular level, isolated *Ucp2^−/−^* macrophages stimulated by lipopolysaccharides (LPS) significantly increased ROS production and synergistically potentiated the regulation of the nuclear factor kappa light chain enhancer of activated B cells (NFκB) and Mitogen-activated protein kinases (MAPK) signaling pathways to enhance the inflammatory response compared to wild-type macrophages [23,99]. These in vitro studies were also confirmed after Leishmaniasis infections. Ball et al. showed in 2011 that mice injected with a *Ucp2* shRNA into the spleen had a parasite load inversely correlated with ROS production [100]. This result was also confirmed in *Ucp2^−/−^* mice [101]. Moreover, in cell lysates from Leishmania-infected *Ucp2^−/−^* macrophages, the increased production of pro-inflammatory cytokines (IL1β, IL6, TNFα) was induced by the activation of Erk and p38 MAPK signaling pathways and the inflammasome [100,102]. UCP2 indeed regulates caspase 1 via the NLRP3 inflammasome by stimulating lipid synthesis, whether during localized or systemic infection [103]. Clinical studies have also corroborated these data by showing a positive correlation between UCP2 expression and the severity of sepsis in patients [104].

#### 2.3.3. Autoimmune Diseases

In autoimmune diabetes (type 1 diabetes (T1D)), the role of UCP2 has also been discussed. Indeed, using the streptozotocin (STZ) model with repeated injections of low doses of STZ that induce selective necrosis of pancreatic β-cells, Emre et al. showed in *Ucp2^−/−^* mice that the development of the disease was advanced and more severe with increased hyperglycemia compared to *Ucp2^+/+^* mice [29]. This phenotype is accompanied by an increased infiltration of pro-inflammatory lymphocytes and macrophages, as well as the increased production of ROS. Indeed, the promotion of inflammation and oxidative stress accelerated the establishment of T1D by causing the rapid destruction of *Ucp2^−/−^* pancreatic β-cells [29]. In contrast, Lee et al. found that the STZ use associated with UCP2 loss essentially alters the function of glucagon-secreting α-cells by increasing ROS [105]. Overall, these results indicate that the loss of UCP2 in autoimmune diabetes can affect both pancreatic β- and α-cells, depending on the study setting and mouse strain (C57BL/6J and C57BL6/129, respectively).

Multiple sclerosis and its corresponding animal model (experimental autoimmune encephalomyelitis (EAE)) is a demyelinating pathology of the central nervous system (CNS). Since no curative treatment has been developed, teams have been interested in studying the impact of the loss of UCP2 on this neurodegenerative disease. *Ucp2^−/−^* mice, immunized with a myelin-specific antigen combined with an adjuvant, developed more severe clinical scores compared to wild-type mice [54]. Greater T cell infiltration was found in *Ucp2*-deficient mice compared with wild-type mice. This accumulation of CD4 and CD8 lymphocytes promoted a strong pro-inflammatory response with the production of high levels of TNFα, IL2 and ROS [54]. Interestingly, the double deficiency of UCP2 and iNOS (inducible nitric oxide synthase) reduced the inflammatory phenotype [106]. More recently, in the spinal cord of wild-type mice, it was shown that the peak of UCP2 expression coincided with the peak of clinical symptoms of EAE occurring 14 days after immunization [107]. These results suggest that UCP2 may be upregulated in the spinal cord to counteract inflammation-induced oxidative stress, and thereby protect the CNS. Accordingly, Alves Guerra et al. showed in 2003 that UCP2 was strongly regulated during the immune response. In the early stages of LPS stimulation, UCP2 expression was inhibited to promote the establishment of the inflammatory response through ROS production. Then, at a later stage, UCP2 was overexpressed to counteract the toxic effect of oxidative stress [25].

In conclusion, all the studies cited above have demonstrated that the mitochondrial transporter UCP2 is involved in energy metabolism and in the regulation of oxidative stress capable of initiating an immune response (Table 1). As both of these features are important aspects of tumorigenesis, we will discuss in the next section how UCP2 might regulate tumor development.

## 3. UCP2 and Cancer

In recent years, many studies have shown increased UCP2 protein expression in a large number of human tumors compared to adjacent normal tissues (e.g., head and neck, skin, pancreatic, prostate, colon, gallbladder, breast cancer, etc.) [118,119,120,121]. After screening more than 100 colon adenocarcinomas, Horimoto et al. showed 3- to 4-fold higher levels of *Ucp2* mRNA and protein in tumors than in normal tissues [118]. Additionally, UCP2 levels were positively correlated with increased amounts of lipid peroxidation and associated with neoplasia [118]. Naturally, these results lead us to wonder whether increased UCP2 expression is protective or deleterious for tumor development. Thus, in this section, we will attempt to elucidate the role of UCP2 in a tumor context by successively examining its involvement in the oxidative stress, metabolic and immune reprogramming of a tumor.

### 3.1. UCP2: A Double-Edged Fight against ROS

Oxidative stress plays a complex role in cancer development. UCP2, whose antioxidant role has been detailed previously, has been implicated in tumorigenesis by controlling ROS. The initiation of colon cancer by azoxymethane (AOM) treatment induces the formation of more aberrant crypts in *Ucp2^−/−^* mice than in *Ucp2^+/+^* mice [108]. In the absence of UCP2, the increase in lesion number was associated with a lack of protection against oxidative stress and the activation of NFκB-dependent anti-apoptotic genes [108]. Furthermore, Aguilar et al. recently showed an increase in tumor number in *Ucp2^−/−^* mice compared to wild-type mice using two models of colorectal carcinogenesis: AOM-DSS (dextran sodium sulfate) and APC^min^ (Adenomatous polyposis coli) [32]. Loss of UCP2 enhanced colon tumorigenesis by promoting lipid synthesis, which limits the availability of nicotinamide adenine dinucleotide phosphate (NADPH) for glutathione synthesis to buffer oxidative stress [32]. In addition, mutations in the *tumor protein 53* (*TP53*) gene occur during tumorigenesis in over 50% of human cancers, and particularly affect colorectal cancer. The mutant p53 protein loses its antioxidant capacity but acquires new pro-tumor, pro-inflammatory and pro-oxidant biological properties [122]. Indeed, the mutant p53 protein inhibits the Sestrin 1 (SESN1)/AMPK/PPARγ coactivator 1 (PGC1) pathway, which is a transcriptional axis that activates UCP2, and consequently the inhibition of this signaling pathway stimulates O_2_^−^ production in cancer cell lines. In this context, the loss of UCP2 associated with the gain of function of the p53 mutant confers hyperproliferative anti-apoptotic effects on cancer cells [123].

However, depending on the stage and type of cancer, UCP2 has a dual regulation mechanism. The hypothesis today is that UCP2 would be repressed in the tumor initiation phases to allow the accumulation of ROS, as described above, while on the contrary the protein would be overexpressed in the late phase to maintain the proliferative capacities of the cells in the tumor microenvironment (TME) that are already under stress conditions [120,124]. This hypothesis is also based on the binary action of ROS. Indeed, when ROS levels are elevated but not uncontrolled, they favor tumor promotion through the acquisition of mutations in oncogenes and the activation of pro-tumor metabolic signals associated with the disruption of the establishment of antioxidant mechanisms. Moreover, a significant increase in oxidative damage will trigger programmed cell death, which relies on caspase activity and cell elimination by phagocytes to limit tumor development. However, tumor cells set up antioxidant activities essential for tumorigenesis to counteract ROS-induced apoptosis [125,126]. Several studies have shown the ROS control induced by UCP2 in established tumor cell lines. Under hypoxic conditions, similar to tumor core conditions, the overexpression of UCP2 in the lung adenocarcinoma cell line A549 showed anti-apoptotic properties. Indeed, the UCP2-induced decrease in ROS accumulation inhibited programmed cell death by blocking cytochrome c release and reducing caspase 9 activity [109]. Correlatively, the transfection of *Ucp2* siRNA into A549 and PaCa44 (human pancreatic adenocarcinoma) cell lines inhibited cell growth through a strong increase in ROS production [109,110]. The loss of UCP2 in different human pancreatic adenocarcinoma cell lines further stimulated autophagy through the ROS-dependent nuclear translocation of glyceraldehyde-3-phosphate dehydrogenase (GAPDH) [110]. Autophagy-derived apoptosis was reversed by treatment with the free radical scavenger N-acetylcysteine (NAC) when *Ucp2* was inhibited genetically by siRNA and pharmacologically by Genipin [110]. Additionally, Wu et al. demonstrated that the silencing of UCP2 in human glioma cell lines alters the p38 MAPK signaling cascade, which reduces the migration and invasion capabilities of tumor cells [127]. Indeed, p38, widely studied as an antitumorigenic factor in the early stages of the disease, has rather an oncogenic role in advanced disease, since it is involved in the metastatic process when it is no longer blocked by dual-specificity phosphatases (DUSP) [127,128]. Therefore, the loss of UCP2 may favor tumor cells that are less likely to metastasize.

### 3.2. UCP2 and Tumoral Metabolic Reprogramming

Metabolic dysregulation is one of the factors identified by Hanahan and Weinberg as a common feature of cancer cells [3]. Moreover, we have seen previously that UCP2 plays an important role in controlling cell metabolism by modulating the TCA cycle activity, but what about under conditions of tumorigenesis?

#### 3.2.1. Glycolysis

Glycolysis is the metabolic pathway that converts glucose to pyruvate. In the absence or presence of oxygen, tumors promote the production of lactate from pyruvate [129]. Tumors have an intensely active metabolism to provide energy and building blocks for unregulated cell proliferation. Thus, due to the dysregulated activation of oncogenes that induce the overexpression of glycolysis-stimulating enzymes, cancer cells oxidize abundant amounts of glucose [8]. In this context, Esteves et al. demonstrated that UCP2 overexpression in the murine melanoma cell line B16F10 induces hypoxia-inducible factor-2α (HIF2α)/AMPK axis-dependent metabolic reprogramming [111]. The authors revealed that the reduced level of fumarate observed in UCP2^high^ cells would be the link between AMPK activation and decreased HIF2α expression [111]. Indeed, a decrease in fumarate contributes to stabilize the enzyme pyruvate dehydrogenase (PDH) and inhibits HIF2α, and would therefore be indirectly responsible for the downregulation of the expression of the glycolytic enzymes hexokinase 2 (HK2) and pyruvate kinase M2 (PKM2) [111,130]. In addition, several studies have demonstrated a positive effect of fumarate on AMPK activity [131,132]. Besides this, the decrease in fumarate accumulation also suggests a better regulation of TCA cycle through the export of C4 metabolites by UCP2 [111]. Hence, UCP2 shifts the glycolysis-dependent metabolism to oxidative phosphorylation, which significantly decreases the proliferation of B16F10 cells and thus reduces the cells’ tumorigenic capacity. These results are consistent with a recently published study indicating that saturation of the mitochondrial malate shuttle, which uses a nicotinamide adenine dinucleotide (NADH) cofactor to reduce oxaloacetate to malate, redirects proliferative cellular metabolism to aerobic glycolysis [133]. To go further, metabolomics fluxes using C^13^-labeled glucose caused an 18% reduction in glucose consumption by the UCP2-overexpressing JB6 P+ murine skin epidermal cell line [134]. In contrast, pyruvate was found to be much more enriched in C^13^ under aerobic conditions in the same murine cell line. A significant percentage of carbons were rewired to the TCA cycle and not to lactate production, to allow amino acid and nucleic acid synthesis [134]. In addition, palmitate oxidation, as measured by the Seahorse XF apparatus, was higher in JB6 P+ cells transfected with UCP2 compared to control cells [115]. These data demonstrate that the presence of UCP2 induces an increase in β-oxidation in a tumor context as well as in a physiological context, as previously shown by Pecqueur et al. [26,115]. However, although the TCA cycle function is enhanced by UCP2 content, the authors do not show whether this correlates with reduced tumor cell proliferation, migration or invasion.

Nevertheless, other studies have not corroborated these initial results. Yu et al. demonstrated a poor prognosis associated with increased expression of *Ucp2* RNA and UCP2 protein in patients with cholangiocarcinoma. Metabolically, the cellular knockdown of UCP2 in the intrahepatic cholangiocarcinoma cell line HuCCT1 and in the extrahepatic cholangiocarcinoma cell line TFK-1 resulted in the attenuation of glycolysis through the activation of AMPK [112]. Phosphorylated AMPK expression, an indicator of its activation, was upregulated in the absence of UCP2 following increased mtROS production and decreased antioxidant activities [111,112]. Indeed, AMPK is an essential sensor for the maintenance of cellular energy homeostasis that is activated by phosphorylation during metabolic stress, such as ROS-mediated oxidative stress [135]. Moreover, activated AMPK reversed the mesenchymal phenotype by inhibiting Akt signaling in cholangiocarcinoma cell lines with low UCP2 expression. Thus, cancer cells lose their ability to migrate and invade healthy tissue [112]. Through its transport activity, UCP2 modified the utilization of substrate by cancer cells. These changes induce Akt-mTOR (mammalian target of rapamycin) signaling, which activates and mediates a glycolytic flux by maintaining an elevated expression of glucose transporter 1 (GLUT1), 6-phosphofructo-2-kinase (PFKB2), and PKM2, which triggers increased lactate release [113,136]. Furthermore, the inhibition of UCP2, either pharmacologically with Genipin or genetically with siRNAs, was sufficient to reverse the metabolic phenotype and reduce proliferation in PaCa44 and JB6 P+ cancer cell lines [113,136].

Although all these results indicate that UCP2 is involved in the glycolytic metabolism of cancer cells, some data are contradictory. These discrepancies can be explained by the fact that the authors used different cell lines that do not necessarily rely on the same energy metabolism. Moreover, the cancer cells used do not express identical basal levels to UCP2.

#### 3.2.2. Glutaminolysis

However, glucose is not the only energy-providing substrate, and glutamine is an important source of nitrogen and carbon for the TCA cycle in tumor cells. Indeed, oncogenes support glutaminolysis to drive mitochondrial metabolism that regulates cellular precursors essential to support increased cell proliferation [137]. The translation of *Ucp2* being partly regulated by glutamine, and UCP2 facilitating TCA cycle function, several teams have analyzed the role of the protein in glutamine dependence in oxidative tumors [22,77]. A positive correlation between the expression level of UCP2 and glutamine dependence has been demonstrated in several human and murine tumor cell lines [114,138]. Indeed, when N18TG2 neuroblastoma cells were deprived of glucose, UCP2 was upregulated in response to glutamine consumption, which became the main source of oxidative phosphorylation [138]. However, Sancerni et al. recently demonstrated that the metabolic adaptations of cells to intra- and extracellular glutamine depend on the basal level of UCP2 [114]. Indeed, the loss of UCP2 in HPB-ALL leukemic cells, which highly express UCP2 and are metabolically dependent on glutamine, decreased their oxygen consumption and redirected their metabolism to glycolysis. This cellular metabolic adaptation induced a decrease in cell proliferation. In contrast, Jurkat leukemia cells, which express low levels of UCP2 and are dependent on glycolytic metabolism, were less metabolically affected by *Ucp2* knockdown. These results show that Jurkat cells have a more flexible and less oxidation-dependent glutamine metabolism than HPB-ALL cells. However, the loss of UCP2 in Jurkat cells further decreased proliferation by altering the biosynthesis of lipid, protein and nucleotide precursors [114]. In addition, to counteract nutritional stress due to glutamine deficiency, the murine neuroblastoma cell line N18TG2 promoted lactate synthesis; nevertheless, in a manner dependent on activating transcription factor 4 (ATF4), the cells entered quiescent-metabolic and -proliferative phases [138]. Raho et al. also demonstrated that in human pancreatic ductal adenocarcinoma (PDAC), the loss of the UCP2 transporter reduced the mitochondrial glutamine catabolism, leading to increased ROS levels and a shortage of aspartate required for protein and nucleotide biosynthesis [34]. Interestingly, the in vitro and in vivo silencing of UCP2 reduced the rate of cell proliferation only in PDACs mutated for Kirsten Rat Sarcoma Virus (KRAS), which is an oncogene that rewires glutamine consumption for the synthesis of aspartate and oxaloacetate. In parallel, aspartate or GSH supplementation partially or totally rescued the cell growth defect observed in the absence of UCP2 [34,139]. Altogether, these results highlight that UCP2 is essential for glutamine-dependent tumors.

#### 3.2.3. Ca^2+^ Signaling

Ca^2+^ signaling is essential for cellular physiological functions, especially in cancer cells, in order to cope with high energy demands and rewire Ca^2+^ requirements [140]. Upon cellular stress, such as increased oxidative stress reflected by elevated H_2_O_2_ levels, receptor tyrosine kinases activate phospholipase Cγ-1 (PLCγ-1), which then cleaves membrane phospholipids, generating two second messengers that stimulate intracellular Ca^2+^ entry [141]. The overexpression of UCP2 in JB6 P+ cells stimulated with the tumor inducer TPA (12-O-tetradecanoylphorbol 13-acetate) enhanced calcium signaling induced by PLCγ-1 upregulation [115]. Cytoplasmic Ca^2+^ is then captured by the endoplasmic reticulum (ER) and transferred via mitochondria-associated membranes (MAMs) to mitochondria, stimulating ETC activity and thereby promoting ATP production [140]. Above, we have already pointed out that UCP2 is required for mitochondrial Ca^2+^ uptake to counteract the inhibitory methylation of the MCU transporter by PRMT1 [61]. Furthermore, it has also been shown that the combined overexpression of UCP2 and PRMT1 improved mitochondrial respiration and cell viability in lung carcinoma cells [142]. In agreement, lung cancer patients with combined *Ucp2*^high^ and *Prmt1*^high^ RNA expression had reduced 5-year survival [142]. However, the prolonged binding between mitochondria and MAMs promotes the risk of mitochondrial Ca^2+^ overload, which, following mitochondrial dysfunction, induces apoptotic cell death. Thus, it has been shown that UCP2 expression is downregulated to avoid overload when the mitochondria–ER interaction is stabilized [116]. Calcium signaling is therefore a finely regulated process that depends on the balance between mitochondria-ER binding and mitochondrial Ca^2+^ uptake via variations in UCP2 expression to optimally promote tumor development.

Altogether, the studies demonstrated that UCP2 acts differently depending on the metabolic dependence of tumors. In conclusion, the energy metabolism of cancers being very different, it is therefore important to characterize precisely their specific metabolism in order to determine whether a positive or negative targeting of UCP2 will be effective against tumor development.

### 3.3. UCP2-Activated Antitumor Immunity

Immunometabolism represents a new area of intense research to find new targets to fight cancer, as the infiltration of immune cells into the tumor is indeed influenced by the metabolic characteristics of the TME [143]. UCP2, through its regulation of ROS production, acts on the inflammatory and immune responses of various pathologies (see Section 2.3). In addition, UCP2 modulates the metabolic demands of cancer cells and thus appears to be an interesting candidate to study the immune reprogramming of tumors. Bioinformatics analyses of the Cancer Genome Atlas (TCGA) database of breast cancer and melanoma patients showed a positive correlation between UCP2 expression and antitumor immune infiltration [117,144]. A better prognosis associated with high UCP2 expression also confirmed these correlative findings [117,144]. Based on these observations, Cheng et al. aimed to decipher the role of UCP2 in the immune response of melanoma tumors. First, transcriptomic studies demonstrated that UCP2 induced an CD8^+^ T cell and conventional dendritic cell (cDC1) infiltration profile [117]. Then, by establishing xenografts with B16-OVA and YUMM1 melanoma cell lines with doxycycline-inducible UCP2 expression, they demonstrated that UCP2 overexpression in these tumor cells induced an immune environment similar to that of melanoma patients. UCP2 facilitated the recruitment of CD8^+^ T cells through an increased release of the chemokine CXCL10 from cCD1, combined with a decrease in the expression of pro-tumorigenic factors (IL10, M-CSF (macrophage colony-stimulating factor) and VEGF (Vascular Endothelial Growth Factor)) and a normalization of the vasculature that amplifies the antitumor infiltration. Moreover, the induction of UCP2 in combination with immunotherapy (anti-PD1 antibody) resulted in reduced tumor volume and improved survival compared with anti-PD1 antibody alone. These results highlight the potent role of UCP2 in increasing the response to immunotherapy treatment [117]. Based on the results of Esteves et al., the authors hypothesized that increased UCP2 expression in melanoma cells limits the aerobic glycolysis of tumor cells, allowing glucose to be available for consumption by lymphocytes [111,117]. Indeed, when lymphocytes are activated and recruited to the tumor site, they essentially consume glucose to release cytotoxic factors [143]. Other authors have shown that UCP2 improves immune recruitment by acting directly on lymphocytes expressing high levels of UCP2. Indeed, the antigenic stimulation of isolated naive CD4^+^ and CD8^+^ T cells induced an increase in UCP2 expression after 24 h, and even more so after restimulation [145]. Furthermore, activated UCP2 inhibits ROS generation to limit ROS-induced apoptosis, and simultaneously regulates glycolytic flux and TCA cycle activity to promote lymphocyte clonal expansion [146].

In conclusion, despite the different functions of UCP2, which may have opposite effects depending on the type and stage of cancer, the authors are in global agreement that the UCP2 protein does not intervene by modulating proton leakage and thus ETC uncoupling. An increase in mitochondrial respiration in the presence of UCP2 independently of the uncoupling function has nevertheless sometimes been described. However, if the effect on metabolism has been well characterized, the effect on tumor immunity remains little studied to date (Figure 2 and Table 1).

### 3.4. Drug Sensitivity and Therapeutic Improvement

Chemotherapy is a very common strategy used to control tumor growth. Antineoplastic drugs, often given in combination with other treatments, specifically or otherwise target the cell cycle to damage DNA and induce apoptosis in tumor cells. However, the effectiveness of chemotherapy is a major challenge. Indeed, many patients fail to respond due to specific mutations or acquire resistance by adapting antioxidant defense mechanisms and altering the TME [148,149]. Through the roles of UCP2, studies have characterized the involvement of UCP2 in the regulation of tumor cell sensitivity to cancer therapies. In patients with serous ovarian carcinoma, one study showed a negative correlation between UCP2 protein expression and platinum sensitivity. Thus, patients with low UCP2 expression had improved overall survival with platinum-based therapy [150]. However, this work was performed using an AbCam antibody, which was found to be unreliable and non-specific by Aguilar et al. [32]. In vitro, cells overexpressing UCP2 induced chemoresistance by modulating ROS that abrogate p53-induced apoptosis [151]. UCP2 also promoted the expression of the mitochondrial detoxifier superoxide dismutase 2 (SOD2) [121]. The pro-survival activity of UCP2 is also mediated by the NFκB/β-catenin axis. After the treatment of gallbladder cancer cells (G-415) with gentamicin, these transcription factors were activated and, in combination with UCP2, promoted cancer cell survival during chemotherapy [121]. Since gentamycin alone induces *Ucp2* gene expression in pancreatic adenocarcinoma cell lines, it has been hypothesized that UCP2 is directly involved in acquired cancer resistance to gentamycin [152]. Conversely, enhanced sensitivity to staurosporine has been reported in several cell lines overexpressing UCP2, reflected in increased pro-apoptotic caspase activities [111]. As pointed out by Wang and colleagues, the deletion of UCP2 under hypoxic conditions increased ROS, but contributed to the stabilization of nuclear factor erythroid 2–related factor 2 (Nrf2), which upregulated the mediate efflux transporter ATP-binding cassette super-family G member 2 (ABCG2), resulting in chemoresistance to cisplatin and docetaxel [153]. Furthermore, these results are consistent with a clinical study that showed poorer response to and survival following cisplatin-based chemotherapy in advanced lung cancer patients with low UCP2 expression [154]. Overall, these studies showed that regulating UCP2 expression, either positively or negatively depending on the type of cancer, represents a therapeutic strategy to be considered to improve responses to chemotherapy (Figure 3).

#### 3.4.1. Genipin

Genipin is a plant chemical extracted from the fruit of Gardenia Jasminoides Ellis. This molecule is widely used in traditional Chinese medicine for its anti-inflammatory, antioxidant, and antipyretic properties. Chemically, Genipin is an excellent cross-linking agent that has been commonly used in industry to produce biomaterials. In 2006, Genipin was also identified as an inhibitor of UCP2; indeed, a specific inhibition of proton transport by the binding of arginine residues on UCP2 was proposed [71]. However, the lack of UCP2-mediated uncoupling highlighted in the studies cited above does not allow us to determine the true mechanism of action of this molecule on UCP2, although the antitumor role of Genipin is well documented [71,155,156,157]. Furthermore, Vozza and his team pointed out in 2014 that Genipin had no effect on UCP2 transport activity. Indeed, this molecule did not catalyze the ^33^Pi/Pi exchange [22]. In contrast, Genipin, by decreasing UCP2 expression in MCF7 breast cancer lines, inhibited tumor promotion characterized by lower cell proliferation and migration compared to control [158]. The growth, migration and 3D steroid invasion of cholangiocarcinoma cancer lines were also effectively reduced by Genipin in a dose-dependent manner [112]. In vitro, Genipin treatment activated intrinsic apoptotic pathways that enabled cell cycle arrest in the G2/M phase through the inhibitory induction of cyclin-dependent kinase inhibitor 1 (p21) [159,160]. Moreover, the triggering of p38 MAPK signaling inducing the pro-apoptotic activity of caspase 3, which enhances the antitumor effect of Genipin [161], appears to be ROS-dependent [110,113]. Indeed, the inhibition of UCP2 by Genipin induced high levels of total ROS measured by the non-fluorescent diacetylated 2′,7′-dichlorofluorescein (DCFH-DA) probe in the cell medium of pancreatic adenocarcinoma lines triggering cell apoptosis through an autophagic process that was impaired by NAC treatment [110]. In addition, Cho et al. recently demonstrated that Genipin supplementation suppressed the uptake of ^18^F-FDG (18Fluorine-fluorodeoxyglucose), a glucose analog and radiotracer, in positron emission tomography (PET) scans of breast (T47D and MDA-MB-435) and colon (HCT116 and HT29) cancer cells [162]. The resulting decrease in lactate production and glycolytic flux has been characterized at the molecular level by inhibition of the expression of glycolytic enzymes (GLUT1, PFKFB2, PKM2, Lactate dehydrogenase a (LDHa)…) associated with Akt signaling [113,136]. Finally, the pleiotropic action of Genipin also induced tumor suppression in vivo by depleting tumor-associated macrophages in the TME of hepatocellular carcinoma mouse xenografts [163].

Nevertheless, the real benefit associated with Genipin treatment appears to be the significant improvement in the sensitivity of cancer cells to antitumor treatments. The combined use of cytotoxic molecules, in this case trastuzumab, with Genipin synergized their action and created a similar effect on the inhibition of cell viability to the use of *Ucp2* siRNA and trastuzumab on the HER2-positive BT474 cell line [164]. In pancreatic adenocarcinoma cells, the combination treatment of Everolimus, an antitumor agent that inhibits the mTOR pathway, with Genipin synergistically inhibited cancer cell proliferation by increasing the nuclear translocation of GAPDH [147]. In pancreatic, lung and leukemia cancer cells, GAPDH is indeed translocated by increased production of ROS, mainly from the mitochondria [147,165,166]. The resulting amplified triggering of apoptosis strongly reduced the tumor volume of pancreatic adenocarcinoma xenografts in mice without altering their general condition during Everolimus–Genipin treatment [147]. Furthermore, Lee et al. also demonstrated that co-treatment with Genipin and Elescomol, a pro-apoptotic chemotherapy adjuvant, decreased tumor energy metabolism by reducing glycolytic flux [165].

However, although Genipin appears to be a promising molecule for improving response to chemotherapy treatments, the lack of knowledge about its molecular mechanism of action has so far prevented the establishment of clinical trials in cancer.

#### 3.4.2. Rosiglitazone

Since the potent role of UCP2-mediated metabolic reprogramming associated with cancer cell proliferation and immunity has been described above in different studies [32,34,111,114,117], UCP2 activators may represent a promising therapeutic strategy to simultaneous target different energetic pathways essential for tumor growth.

After activation, PPARs, members of the steroid receptor family, are translocated into the nucleus to heterodimerize with the retinoid receptor (RXR) and thus initiate the transcription of target genes such as UCP2 [167]. Among the three PPAR isoforms (PPARα, PPARβ/δ, PPARγ), PPARγ is the most studied to date due to its crucial role in carbohydrate and lipid homeostasis, regulation of apoptosis and tumor progression [168,169]. Numerous studies have demonstrated that PPARγ acts as a tumor suppressor by activating the pro-apoptotic gene *TP53* and blocking the cell cycle through the overexpression of p21 in colon, lung, pancreatic, prostate and breast cancer cell lines [170,171,172,173,174]. Furthermore, Kwon and colleagues showed that reduced levels of PPARγ in sporadic colorectal cancers correlated with poor prognosis in patients [175]. The search for synthetic activating molecules led to the identification of thiazolidinediones, including rosiglitazone, as PPARγ agonists [168]. Thus, rosiglitazone has been shown to transcriptionally induce *Ucp2* [176]. Recently, increased UCP2 expression following rosiglitazone treatment in melanoma cell lines promoted antitumor immunity and induced the reprogramming of the cytokine profile secreted by tumor-infiltrating T cells [117]. UCP2 induction also significantly improved the response to anti-PD1 immunotherapy, and a greater reduction in tumor volume in melanoma xenografts was reported when anti-PD1 was combined with rosiglitazone [117]. Rosiglitazone also improved radiosensitivity and response to 5-fluorouracil chemotherapy by inhibiting cell growth and suppressing pancreatic and colorectal cancer cells in vitro [177,178].

However, these potential therapeutic benefits against cancer need to be re-evaluated in humans in terms of dose, because, unfortunately, serious side effects in patients such as fluid retention, heart failure or an increased risk of developing bladder cancer have been described [179]. Although rosiglitazone is still approved by the Food and Drug Administration (FDA), its regulation and use are now very restricted [180]. Moreover, PPAR agonists activate a multitude of target genes, so identifying a specific activator of UCP2 remains a challenge.

## 4. Conclusions

Cancer can be initiated by an uncontrolled accumulation of ROS, and then cancer cells survive and proliferate until they invade other organs by metabolic and immune reprogramming. Although complex to demonstrate, UCP2 is involved in all these steps with different consequences. Its chemical role has been widely reconsidered in recent years following the questioning of its uncoupling function. In this review, we described that UCP2 acts as a metabolic sensor through its role as a mitochondrial transporter of C4 metabolites of the TCA cycle to regulate and control these processes. Overall, we argue that UCP2 transport-related regulations occur not only in a tumor context, but also in a healthy context. Metabolite modulation then leads to adaptations in cellular and mitochondrial metabolism. Nevertheless, we believe that the metabolic adaptations enabled by UCP2 are more pronounced in a stressful context, such as cancer. Indeed, few effects due to a modification of UCP2 expression can be observed in a model of healthy intestinal epithelium, for example. The latter theoretically has the capacity to maintain its ATP production through the different pathways of energy catabolism. However, when cells are damaged in metabolic diseases, infections or cancers, a loss of UCP2 will have more drastic consequences. Cancer cells or immune cells are indeed dependent on specific metabolic pathways, such as glutamine, which are essential for their activation and expansion. Thus, the absence of UCP2 will lead to a saturation of their TCA cycle, totally unbalancing the metabolism of these cells, and this will decrease in parallel the cellular antioxidant defenses caused by the absence of aspartate transport, and consequently of glutathione synthesis.

In clinical studies, different correlations, negative or positive depending on the type of cancer, have been described between *Ucp2* mRNA expression and patient survival. UCP2 is therefore considered today in many studies based on the Oncomine database as a potential cancer biomarker. However, it is crucial to remember that *Ucp2* gene expression is not correlated with protein levels due to translational regulation [19]. Moreover, many commercial antibodies targeting UCP2 commonly used by different teams are not effective, and are sometimes even non-specific, because the detected signal is conserved in *Ucp2^−/−^* tissues [32]. Thus, caution should be taken in correlating the prognosis of cancer progression with UCP2 protein levels in cancer patients, as has been done in breast, pancreatic and lung cancer with non-specific antibodies [33,124,142]. Nevertheless, based on all the studies cited above, the inhibition or activation of UCP2 according to the metabolic status of the tumor represents a therapeutic opportunity to improve cancer remission. Targeting UCP2 in combination with conventional treatments (chemotherapy, radiotherapy, immunotherapy) could promote a beneficial response in patients. However, we have seen here that the use of Genipin or rosiglitazone is not optimal because their mode of action on UCP2 is not well characterized and not specific. Designing or synthesizing new molecules directly targeting UCP2 should be a priority in the coming years to counter tumor progression.

## Figures and Tables

**Figure 1 ijms-23-15077-f001:**
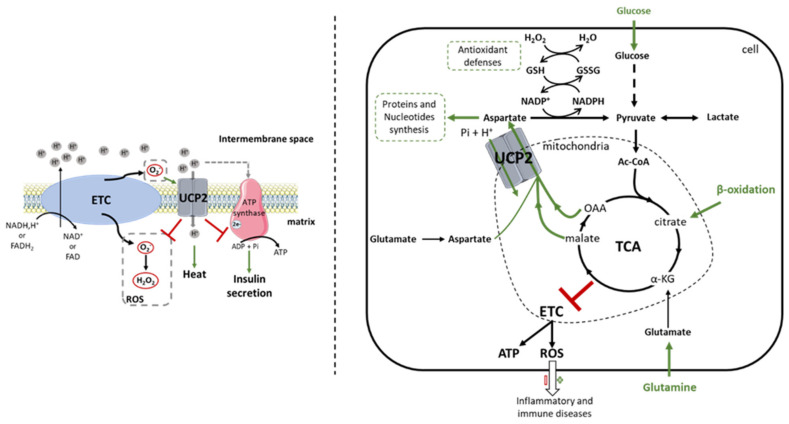
Comparison between the first function attributed to UCP2 and the new concept of mitochondrial transport highlighted by the work of the Fiermonte team. Since its discovery, the UCP2 protein has been the subject of much controversy regarding its biochemical function. UCP2 was initially identified as an uncoupling protein of the electron transport chain. Thus, the protein captures protons from the intermembrane space and dissipates them into the mitochondrial matrix. Increased thermogenesis, decreased reactive oxygen species (ROS) levels and ATP synthesis, which consequently reduce insulin secretion, are observed due to reduced membrane potential (**left**). However, the non-reproducibility of these results in different cells and tissues has questioned this function. Since then, work by Vozza et al. [22] in 2014 showed that UCP2 catalyzes the exchange of 4-carbon (C4) metabolites (oxaloacetate, malate, aspartate) from the tricarboxylic acid (TCA) cycle for a phosphate and a proton. UCP2 then regulates pyruvate oxidation in the mitochondrial matrix, which promotes cell proliferation by providing building blocks and increases antioxidant defenses by regulating glutathione oxidation. C4 transport also controls TCA cycle activity to avoid overload by the accumulation of intramitochondrial compounds. The redox pressure of the oxidative phosphorylation is then attenuated, which limits the production of ROS more or less harmful in the fight against pathologies (**right**). Ac-CoA: Acetyl-CoA; ADP: adenosine diphosphate; ATP: adenosine triphosphate; C4: 4-carbon metabolites; ETC: electron transport chain; FADH2(-FAD): flavine adenine dinucleotide; GSH(-SSG): glutathione; H^+^: hydrogen ion; H_2_O: water molecule; H_2_O_2_: hydrogen peroxide; NADH(-NAD^+^): nicotinamide adenine dinucleotide; NADPH(-P^+^): nicotinamide adenine dinucleotide phosphate; O_2_-: superoxide; OAA: oxaloacetate; Pi: phosphate ion; ROS: reactive oxygen species; TCA: tricarboxylic acid cycle; UCP2: uncoupling protein 2.

**Figure 2 ijms-23-15077-f002:**
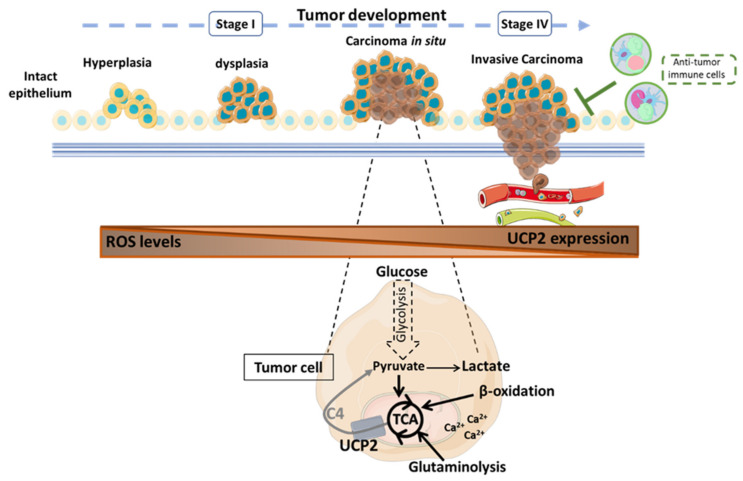
Role of the mitochondrial carrier UCP2 in tumor progression according to published results [32,111,114,116,117,147]. UCP2 is involved at all stages of tumor development with different consequences depending on the type of cancer. UCP2 acts as an antioxidant to counteract the formation of cellular and mitochondrial reactive oxygen species that could then spread into the tumor microenvironment (TME) to promote the acquisition of new pro-tumor properties of the cells. Through the transport of 4-carbon metabolites of the tricarboxylic acid (TCA) cycle, UCP2 is involved in the regulation of tumor metabolism. When UCP2 is expressed, tumor cells undergo oxidative metabolic reprogramming. Thus, glucose is preferentially converted to pyruvate to support cellular anabolism through TCA cycle activity. In addition, the export of C4 metabolites prevents the overloading of the TCA cycle via the accumulation of intramitochondrial compounds and reduces redox pressure to limit ROS production. Moreover, β-oxidation and glutaminolysis can be promoted to sustain TCA. Calcium dependence also supports the energy requirements of tumor cells. Nevertheless, the presence of UCP2 correlates with a significant antitumor immune infiltration to resist tumor development. C4: 4-carbon metabolites; Ca^2+^: calcium ion; ROS: reactive oxygen species; TCA: tricarboxylic acid cycle; UCP2: uncoupling protein 2.

**Figure 3 ijms-23-15077-f003:**
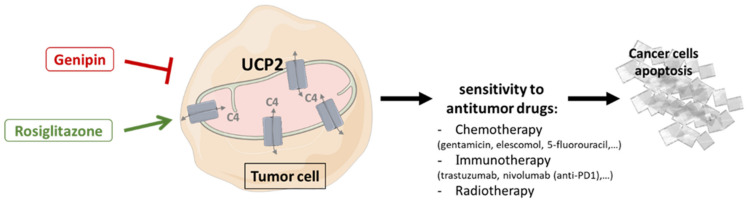
Therapeutic strategies targeting UCP2 combined with antitumor therapies to promote cancer cell apoptosis. Use of activators (rosiglitazone) or inhibitors (Genipin) of UCP2, depending on tumor-dependent metabolism, leads to changes in the capacities of mitochondria to provide energy and building blocks through the transport of 4-carbon tricarboxylic acid cycle metabolites, and may improve sensitivity to anticancer therapies by increasing tumor cell apoptosis. C4: 4-carbon metabolites; UCP2: uncoupling protein 2.

**Table 1 ijms-23-15077-t001:** Non exhaustive overview of pathophysiological implications of the mitochondrial protein UCP2.

Disease	Experimental Model	UCP2 Status	Impact	Ref.
**Type 2 diabetes**	ob/ob mice	*Ucp2^−/−^*	Increased glucose-stimulated insulin secretion	[42]
		*Ucp2* siRNA	Oxidative stress imbalance	[91,92,93]
**Atherosclerosis**	Atherogenic diet Ldlr-/-mice	*Ucp2^−/−^* Bone marrow	Increased atherosclerotic lesions Increased invasion of macrophages into the intima Oxidative burst	[30]
**Infections**	Toxoplasma gondii	*Ucp2^−/−^*	Resistance to infection by increased production of ROS and pro- inflammatory molecules	[27]
	Listeria	*Ucp2^−/−^*		[28]
	Leishmaniasis	*Ucp2^−/−^* *Ucp2* shRNA		[100,101]
**Autoimmune diseases**	Streptozotocin (type 1 diabetes)	*Ucp2^−/−^*	Higher disease scores Increased oxidative stress and inflammation	[29]
	Experimental autoimmune encephalomyelitis			[54,106,107]
**Cancer**			**Oxidative stress:**	
	AOM/AOM-DSS / APC^min^ (colorectal cancer)	*Ucp2^−/−^*	Decreased protection against oxidative stress and increased colorectal tumorigenesis	[32,108]
	A549 cell line (lung cancer)	UCP2 overexpression	Reduction of ROS accumulation conferring anti-apoptotic properties	[109]
	A549 and PaCa44 cell line (lung and pancreatic cancer)	*Ucp2* siRNA	ROS stimulate apoptosis derived from autophagy	[109,110]
			**Glycolysis:**	
	B16F10 cell line (melanoma)	UCP2 overexpression	Less tumorigenic cells through down-regulation of glycolytic enzymes	[111]
	HuCCT1, TFK-1 and PaCa44 cell lines (bile duct and pancreatic cancer)	*Ucp2* siRNA	AMPK activation decreases glycolytic activity and therefore the cell invasiveness	[112,113]
			**Glutaminolysis:**	
	HPB-ALL cell line (leukemia)	*Ucp2* CRISPR	Reduction of oxygen consumption Shift of metabolism to glycolysis Low nucleotide synthesis Decreased cell proliferation	[114]
	Patu8988T, Panc1 and BxPC3 cell lines (pancreatic cancer)	*Ucp2* shRNA		[34]
			**Ca^2+^ signaling:**	
	JB6 P+ and HeLa cell lines (skin and cervix cancer)	UCP2 overexpression	Increased calcium activity stimulates ATP production Long-term mitochondrial dysfunction	[115,116]
			**Immune response:**	
	Xenografts in mice with B16-OVA and YUMM1 cell lines (melanoma)	UCP2 overexpression	Better prognosis Infiltration of CD8^+^ T cells and cDC1 cells Improved response to immunotherapy	[117]

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
