# Peer review of "UCP2 as a Cancer Target through Energy Metabolism and Oxidative Stress Control"

_ijms, 2022, doi:10.3390/ijms232315077_

Round 1

Reviewer 1 Report

Luby and Guerra have written a fantastic review on an important subject in cancer biology. Mitochondrial uncoupling regulation is a generally unappreciated topic but its influence over many processes involved in obesity and cancer progression warrants a review like this one. The authors have done a good job linking past research on the mitochondrial uncoupling protein and tumor biology, but as their conclusions state the issue is still far from resolved as highlighted by the seemingly pleiotropic nature of the UCP2 gene. Perhaps the authors could add additional descriptions of the degree to which each of the described cellular functions are carried out by UCP2 (or at least provide informed hypothesis regarding the percentage of each role UCP2 might perform in healthy tissues compared to tumors). Apologies for missing this if this was already included and discussed.

Author Response

Please see the attached pdf file

Reviewer 2 Report

The authors provided a good overview on the role of UCP2 in tumor context. 

I would suggest the authors to provide a table summarizing the main data described in the review (e.g role of UCP2 in different diseases and cancers, effects on tumor cells).  

Author Response

We would like to thank the reviewer for her/his interest in our work and for the insightful comments that helped us to improve our manuscript. Our detailed answers are given hereafter in red.

The authors provided a good overview on the role of UCP2 in tumor context.

Moderate English changes required

English editing has been done. (Please see the attached pdf file)

I would suggest the authors to provide a table summarizing the main data described in the review (e.g role of UCP2 in different diseases and cancers, effects on tumor cells).

We have added Table 1 to summarize the main data described in the review
